# Early Maladaptive Schemas and Their Impact on Parenting: Do Dysfunctional Schemas Pass Generationally?—A Systematic Review

**DOI:** 10.3390/jcm12041263

**Published:** 2023-02-05

**Authors:** Klaudia Sójta, Dominik Strzelecki

**Affiliations:** Department of Affective and Psychotic Disorders, Central Teaching Hospital, Medical University of Łódź, ul. Czechosłowacka 8/10, 92-216 Łódź, Poland

**Keywords:** early maladaptive schemas, intergenerational transmission, parenting

## Abstract

There are several factors that play a key role in the development of early maladaptive schemas, i.e., temperament, unmet core emotional needs, and adverse childhood events (e.g., traumatization and victimization, overindulgence, overprotection). Thus, the parental care that a child experiences has a substantial impact on the potential development of early maladaptive schemas. Negative parenting can range from unconscious neglect to overt abuse. Previous research supports the theoretical concept that there is a clear and close relationship between adverse childhood experiences and the development of early maladaptive schemas. Maternal mental health problems have been proven to be a factor that has strengthened the link between a mother’s history of negative childhood experiences and subsequent negative parenting. Consistent with the theoretical background, early maladaptive schemas are associated with a wide variety of mental health problems. Clear links have been found for EMSs and personality disorders, depression, eating disorders, anxiety disorders, obsessive-compulsive disorder, and post-traumatic stress disorder. In light of these theoretical and clinical connections, we decided to summarize the available literature on the multigenerational transmission of early maladaptive schemas, which is also an introduction to our research project.

## 1. Introduction

From the moment of birth, a child forms beliefs about the surrounding world, themself, and others, which initially come from the relationship with their parents. This relationship plays a crucial role in shaping personality patterns, which has long-term consequences in maintaining mental health in adulthood. Representatives of various theoretical approaches consistently postulate that a safe parent–child relationship, in which the caregiver adequately responds to the child’s core needs, is the basis for proper personality development [1,2]. Conversely, deprivation of these needs in childhood is a significant predictor of mental disorders throughout life [3]. There is robust evidence that malparenting(ranging from unconscious neglect to overt abuse) is a notable psychosocial risk factor for the development of personality disorders, more specifically borderline personality disorder [4]. Young proposed a theoretical concept of early maladaptive schemas (EMSs) that facilitate the development of a wide spectrum of mental problems, especially personality disorders [5]. In line with schema theory, which extended Beck’s original work on schemas, adverse childhood experiences with primary caregivers and repeated failures to meet a child’s basic emotional needs lead to the development of early maladaptive schemas. Young et al. defined EMS as ‘a broad, pervasive theme or a pattern, comprised of memories, emotions, cognitions, and bodily sensations, regarding oneself and one’s relationships with others, developed during childhood or adolescence, elaborated throughout one’s lifetime, and is dysfunctional to a significant degree’ [6]. Originally, 18 schemas grouped into five main domains were identified; however, recently, as a result of factor analysis, the four domains of the higher-order schema were identified as more adequate in respect of interpretability and empirical indices [7]. In the current review, due to the predominance of research based on the original theoretical concept, we refer to the five-domain organization of schemes (as detailed in Table 1).

The origin of the schemas is the result of several factors, i.e., temperament, unmet core emotional needs, and adverse childhood events (e.g., traumatization and victimization, overindulgence, overprotection) [6]. Adverse childhood experiences are commonly defined as childhood events that occur in the family or social environment of a child, of varying severity, often chronic, that force the child to psychologically, socially, and neurodevelopmentally adapt beyond the expected developmental norm [8]. Adverse childhood experiences have a profound impact on lifespan health and well-being. Previous research supports the theoretical concept that there is a clear and close relationship between adverse childhood experiences and the development of early maladaptive schemas [9,10]. In addition, studies have shown that the impact of childhood adverse events on personality disorder development is mediated and sustained by early maladaptive schemas [11,12].

There is also clear evidence of the intergenerational transmission of child abuse [13,14]. It has been repeatedly noticed that children of parents who were exposed to abusive behaviors from their own parents are significantly more likely to experience domestic violence, which is called “perpetuation across generations” [15]. There are two possible pathways of sustaining the cycle of intergenerational violence: homotypic and heterotypic transmission. The first one is when the next generation repeats some type of aggressive behavior it has experienced, for example, a parent who has been physically neglected tends to develop the same negative parenting practice. The second one concerns a situation when the parent develops other forms of child abuse than those from which he/she suffered [16]. The most recent systematic review investigating the effect of negative childhood experiences on motherhood found that these early adversities experienced by mothers significantly increased the later levels of parental stress. In turn, perceived parental stress enhanced the probability of malparenting. Importantly, the mother’s mental health problems were a factor that strengthened the association between the mother’s history of adverse childhood experiences and subsequent negative parenting [17]. Although the relationship between adverse childhood experiences and early maladaptive schemas is well established in research, little is still known about the role of EMS in perpetuating negative parenting across generations.

Consistent with the theoretical background, early maladaptive schemas are associated with a wide variety of mental health problems. Clear links have been found for EMSs and personality disorders [18], depression [19,20], eating disorders [21], anxiety disorders, obsessive-compulsive disorder, and post-traumatic stress disorder [22].

Not surprisingly, in view of these theoretical and clinical links, researchers wanted to examine whether the early maladaptive schemas, closely related with adverse childhood experiences and affecting mental health in adulthood, could pass from one generation to another via parenting practice. Although a decade has passed since the first empirical study investigating these associations, the literature has not yet been formally synthesized.

### The Current Review

The aim of this systematic review was to evaluate the evidence on the intergenerational transmission of EMS, along with the underlying mechanisms. Identifying the relationship between early maladaptive schemas of parents’ and a child’s can facilitate more complete understanding of the etiology of mental health problems and attempt to explain their accumulation in families. The conclusions drawn can be used to formulate therapeutic guidelines for family interventions that may break the intergenerational cycle of psychological problems [23]. Finally, the acquired knowledge may point the way for future research by addressing gaps in the literature and critically evaluating findings to date.

## 2. Methodology

The presented systematic review on the intergenerational transmission of early maladaptive schemas was carried out in compliance with the Preferred Reporting Items for Systematic Reviews and Meta-Analyzes (PRISMA) guidelines [24]. The protocol was registered with the PROSPERO database of systematic reviews (http://www.crd.york.ac.uk/prospero; accessed on 30 August 2022; registration number CRD42022350839). Figure 1 shows the PRISMA flow chart for the screening and selection of studies in conducted research.

### 2.1. Eligibility Criteria

We adhered to the following criteria to decide whether a study was eligible for inclusion: (I) sample of adult participants with a full parental rights of at least one child; (II) analysis of therelationship between EMSs of parents and their children; (III) focusing on associations between early maladaptive schema of parent and various aspects of parenting (e.g., parental attitudes, sense of parenting competences, parenting styles, etc.), taking into account assessments made by the parent–child dyad or the parent;(IV) adaptation of a case–control, longitudinal, cross-sectional, or retrospective study design; (V) published peer-reviewed articles written in English.

### 2.2. Search Strategies

In the first step, we developed a search strategy by specifying keywords and their combinations. We have established the following equation: “Early maladaptive schemas” AND (“Parenting Style” OR “Parental Competence” OR “Parent Attitudes” OR “Attachment style” OR “Bonding”), which was used to run the search in five electronic databases: PubMed, EBSCO, MEDLINE, ScienceDirect, Scopus. We limited the search to peer-reviewed publications of original research written in English up to and including 2022. The literature search was completed on 10 September 2022. Authors checked the databases and collected the data separately. Through discussion and the consent of both researchers, inclusion of the study into the list was proceeded.

### 2.3. Methodological Quality Assessment

We separately assessed the quality of the selected studies using the Newcastle-Ottawa Scale [25], customized for cross-sectional studies by Modesti et al. [26]. For each study, we performed the ratios using the following criteria: (1) Representativeness of the sample: (a) truly representative of the average in the target population (all subjects or random sampling), (b) somewhat representative of the average in the target population (non-random sampling), and (c) unclear or no description of the sampling strategy; (2) Sample size: (a) justified and satisfactory and (b) not justified; (3) Ascertainment of the exposure (risk factor): (a) validated measurement tool, (b) non-validated measurement tool, but the tool is available or described, and (c) no description of the measurement tool; (4) The subjects in different outcome groups are comparable, based on the study design or analysis. Confounding factors are controlled: (a) The study controls for the most important factor (select one), (b) The study control for any additional factor; (5) Assessment of outcome: (a) independent blind assessment, (b) record linkage, (c) self-report, and (d) no description; (6) Statistical test: (a) The statistical test used to analyze the data is clearly described and appropriate, and the measurement of the association is presented, including confidence intervals and the probability level (*p* value). (b) The statistical test is not appropriate, not described, or incomplete.

### 2.4. Main Outcomes

Establishing a relationship between EMSs of parents and the development of EMSs in their children and determining the connections between the different EMSs domains and parenting (e.g., parental attitudes and styles, sense of competence) were our main outcomes. These outcomes had to be measured by standardized and validated scales of early maladaptive schemas (domains, modes, coping styles)—based on Young’s theory of Early Maladaptive Schemas [5,6] and parental styles, attitudes, and attachment.

### 2.5. Data Extraction (Selection and Coding)

We extracted data from the studies that meet full inclusion criteria, using a standardized extraction sheet and codebook. Both authors independently extracted data to minimize the risk of inaccurate extraction. Any discrepancies were resolved through discussion. The following variables were assessed: (a) reference information, (b) basic descriptive information about the sample (sample size, sample characteristic), (c) the objectives, (d) type of measures, (e) results and conclusions.

### 2.6. Data Synthesis

Due to the nature of the research described, a narrative synthesis was used to synthesize the studies included in the review. In the first step, we performed a preliminary synthesis for clear data organization. Extracted data were presented in tabular form. This approach facilitated the visualization of data, the observation of relationships within and between studies, and making comparisons. Simultaneously, we separately created a textual description of each selected study, assigning importance ranks based on methodological quality criteria. We then divided the included studies into groups to explore relationships within and between the selected studies. The studies were grouped according to the following criteria: target group (e.g., parent, parent–child dyad); outcome measures, various aspects of EMS impact (e.g., bonding, parental sense of competence). Finally, we critically considered the synthesis product, with an emphasis on potential limitations and their impact on research results. We have explained any discrepancies and ambiguities during the discussion.

## 3. Results and Discussion

### 3.1. Included Studies

We present summarized characteristics of the selected studies in Table 2. We searched five electronic databases and found 199 references (77 from PubMed, 57 from Scopus, 14 from ScinceDirect, and 51 from EBSCO and Medline). After removing duplicates, we obtained 111 titles, which were then checked for compliance with the inclusion criteria. As a consequence, we excluded 42 references that were not relevant to the topic. Subsequently, we analyzed 69 abstracts, of which we excluded 60 because they did not meet the inclusion criteria. There were two main reasons for exclusion: (I) Retrospective assessment of perceived parental care (relating to the child’s perception of parents) (*n* = 34), (II) Inclusion of adult (not parent to child) attachment tool (*n* = 26). Then, we included 3 manuscripts through a manual reference lists search. As a consequence, we analyzed 12 full manuscripts, and two were excluded due to failure to meet the eligibility criteria (parental EMSs have not been assessed) [27,28]. Finally, we included 10 manuscripts in the systematic review.

### 3.2. Study’s Characteristics

The included articles were published between 2012 [29] and 2022 [38]. The years with the highest number of publications are 2018 (*n* = 2) and 2019 (*n* = 2). The research was conducted on the populations of four continents: Europe [30,31,32,33,35], Asia [36,37,38], North America [29], and Australia [34]. All manuscripts were peer-reviewed journal articles.

### 3.3. Sample’s Characteristics

The sample size ranged between 40 [33] and 626 participants [37]. Most of the studies (*n* = 7) concerned parent–child dyads (e.g., mother–daughter). Two of the studies analyzed included whole families [30,37]. In one study, only parents were assessed [31]. One study investigated the bond that forms between the mother and the unborn fetus [35]. The mean age of the caregivers assessed in the studies varied between 30.8 [35] and 58.13 [29]. The analyzed studies evaluated the impact of parents’ EMSs on two age groups of respondents: adolescents and adults. The populations included in the analyzed studies were of clinical and non-clinicalsamples.

The main objectives of the selected studies were: (I) to examine the interrelationship and differences between EMSs among parents and their children [29,30,32,33]; (II) to evaluate mediating factors between parents’ and children’s EMSs [34,36,37,38]; (III) examining the relationship between parents’ EMS and the characteristic features of their parenting (i.e., perceived parental competence [31], quality of the affective bonding [35].

### 3.4. Measurement Characteristics

All selected studies measured outcomes using standardized and validated self-report instruments of early maladaptive schemas (domains, modes, coping styles), based on Young’s theory of Early Maladaptive Schemas and parental styles, attitudes, and attachment. One study also used a testing tool (i.e., the Autobiographical Memory Recall Task) to evaluate mediating factors between parents’ and children’s EMSs [38]. Most studies have examined the effects of EMSs on parenting by comparing the results of different measures in dyads or families (*n* = 9). A cross-sectional design was used in all of the studies.

### 3.5. Main Findings

#### 3.5.1. Characteristics of the Relationship between Parents’ EMSs and Their Children’s EMSs

The studies conducted so far have not obtained unequivocal support for the hypothesis of direct transmission of early maladaptive schemas between generations. Several of the 18 schemas were consistent in parent–child dyads, with varying size effects for the correlations reported. Only two of the EMSs have been repeatedly identified as consistent in parent–child dyads. The first one mentioned is the Self-sacrifice schema, which is characterized by an excessive focus on the needs of other people at the expense of one’s own [29,30] in order to avoid guilt, to maintain relationships, and to help others not to experience pain [6]. This schema was also described as the most burdensome [33]. The second schema, for which significant positive correlations between parents and children scores were found, was Enmeshment/Undevelopedself [30,34]. The consequence of mentioned schema is excessive emotional involvement in relatives’ lives and a sense of confusion about one’s own identity [6]. Other EMS that were each significantly associated between parent and childinclude: Defectiveness/Shame, Failure, Vulnerability to harm or illness, Emotional deprivation [30], Abandonment, Socialisolation, Subjugation and Approval seeking [34]. However, the results obtained by successive researchers have varied, without revealing a clear pattern of “replication” of EMSs between generations.

There were discrepancies between the researchers about the parental schema, which was associated with the strongest impact on the overall intensity ofthe child’s EMSs. Mącik’s study identified parents’ EMSs differing in terms of gender, which had the most intense impact on the development of EMS in children. For mother–daughter dyads, the mother’s Vulnerability to harm or illness schema was positively correlated with the highest number of increased schema scores (*n* = 15) in daughters. Maternal Insufficient self-control schema, on the other hand, was associated with the highest rates for two of the sons schemas, which was the strongest indicator of the overall intensity of the son’s schemas. As in the case of mother–daughter dyads, the father’s Vulnerability to harm or illness schema turned out to be the strongest predictor of EMS severity in sons. Two schemas of fathers Defectiveness/shame and Subjugation were significantly associated with the highest number of daughter’s schemas (*n* = 8) [30]. Other researchers point to different parental schemas as being most strongly associated with the severity of EMS in their children. Three other parental schemas were found to be the strongest predictors of overall schema expression in children, which were, Entitlement corresponding positively and with large effect size to three child’s schemas [33], Abandonmentand Mistrust/Abuse, both showing positive associations with medium effect size with six EMSs of children [34].

Most of the researchers showed significant connections between the intensity of parent’s EMSs and their development in children, which may constitute preliminary evidence confirming the intergenerational transmission of schemas; however, overwhelmingly, the impact is not direct. Researchers used the hypothesis with the assumption of which schema “inheritance” occurs through the child’s schemas responding to those present in their parents, rather than as a simple duplication of schemas between generations [29,30,33,37]. For example, in the Mącik study, the strongest correlations with the daughter’s EMS were noted for the Vulnerability to harm. The authors postulated that a fearful, tense, and anxious mother, preoccupied with catastrophic scenarios of the future, is not able to provide her daughter with a safe environment that strengthens self-confidence and a sense of agency. As a consequence, the daughter becomes insecure about her own resourcefulness and compliant with others (Dependence/Incompetenceschema) [30]. Zonnevijlle [10], on the other hand, noted that a parent with intense expression of Entitlement/Grandiosity and Insufficient self-control/self-discipline schemas may have trouble taking responsibility for the child, create a chaotic, unpredictable environment, show egocentrism and selfishness, have problems with emotional control, and even be aggressive. As a consequence, the child’s needs for security, stability, predictability, and understandability of the surrounding environment, as well as the willingness to be taken into account, seen, and accepted, cannot be sufficiently met, which in turn creates conditions for the development of the Mistrust/Abuse and Defectiveness schemas. The parenting attitude described above may also lead to the development of child’s beliefs about a dangerousworld, full of catastrophes, towards which the child feel helpless (Vulnerability to harm schema) [33]. While research suggests the hypothesis that children develop early maladaptive schemas in response to parental EMS, the results obtained in the studies are heterogeneous and point to several possible pathways for schema development in children, without identifying any of them as crucial.

According to research, the Disconnection/Rejection domain of parents is consistently shown as the most harmful, threatening the fulfillment of basic needs necessary for the harmonious emotional development of a child. This domain contains five early maladaptive schemas, resulting from the deprivation of core need for safe, stable, predictable family environment that shows respect, appreciation, empathy, love and interest [6]. The Disconnection/Rejection domain is also closely related to phenomena of parental emotional neglect and emotional abuse in childhood [10]. What is more, schemas of the Disconnection/Rejection domain are identified as highly prevalent in BPD populations [39]. Not surprisingly, high parental scores for this domain repeatedly showed the strongest associations with children’s early maladaptive schemas, which is in line with previous research on the continuity between the generations of harsh and hostile or aggressive parenting [13,40,41]. Parents’ Disconnection/Rejection domain schemas had twofold influence: first, they were transferred directly to their children [34,36,38], second: they contributed to the development of EMSs from other domains (Dependence/Incompetence, Failure, Negativity/Pessimism, Grandiosity, Self-sacrifice, Enmeshment/Undeveloped self, Vulnerability to harm or illness, Insufficient self–control, Punitiveness, Subjugation) [30,33,34,37].

In addition, the researchers pointed to two other domains of parental schemas, the severity of which played an important role in the development of schemas in children. The findings so far suggest that there are substantial associations between Impaired Autonomy and performance domains of parents and the Disconnection/Rejection and Impaired Autonomy EMS domains of their children [30,37]. In one study, the severity of parental schemas from the Impaired limits domain was significantly associated with the prevalence of schemas fromDisconnection/Rejection and Impaired autonomy and performance domains in children [33].

Finally, the studies conducted so far do not provide clear conclusions regarding the mechanisms responsible for the direction of intergenerational transmission of early maladaptive schemas. Researchers identified several potential mediators that would determine whether, and if so, how the child will respond to the parents’ early maladaptive schemas. The first study to test the mechanisms explaining the intergenerational “inheritance” of EMS confirmed that the association between EMSs in parents and their offspring might be explained by the extent of adverse parental behaviors that the child recalled. It was assumed that those negative, abusive behaviors of the parent that the child remembered were related to parent’s dysfunctional schema coping style, precisely—Overcompensation. Overcompensation mechanisms are expressed through rigid, unrelenting, excessive control-oriented behavioral patterns to prevent negative emotions arising from EMS. What is worth emphasizing is that this specific effect did not remain significant when Overcompensation occurred, regardless of the child’s memories of adverse parental behaviors. Based on these observations, the researchers hypothesized that it is the way the child perceives and remembers family experiences, and not the parents’ dysfunctional behavior itself, that might be critical for the transmission of EMSs from one generation to the next [32]. This hypothesis may be partially supported by the results of the most recent study included in this review. Alaftaret al. showed that the over-general memory tendency (OGM) may increase the probability of developing EMSs in children after adverse childhood experiences. OGM is associated with cognitive vulnerability to an over-generalized way of recalling events from the past, which may prevent adequate restructuring of negative experience, and further create facilitating conditions for the development of early maladaptive schemas [38]. The impact of children’s perceived negative parental behavior appears to be of overwhelming importance in the development of EMSs in these children, as further research indicates [33,36,38]. A parent who suffers from their own maladaptive schemas may find it difficult to respond adequately to the child’s needs and provide a sufficiently safe, attentive, supportive environment. Drawing on the internalized attitudes of parents who were neglected emotionally, as adult he can often reproduce those adverse behaviors that hurt them during childhood. In line with the theoretical background, such an unfavorable care environment may contribute to the development of similar maladaptive schemas in the next generation. Those findings are also consistent with many other studies showing positive correlations between unfavorable childhood experiences and the occurrence of EMS [10].

One study included in the review found that immature defense styles (acting out, denial, devaluation, displacement, dissociation, autistic fantasy, isolation, passive aggression, projection, rationalization, somatization, and splitting) had an indirect, mediatory effect on the relationship between parents’ EMSs and their adult children’s EMSs. Immature defense styles can significantly distort the image of reality, and thus foster false representations of one’s own or one’s children. This, in turn, can lead to inadequate, over-controlling or overprotective parenting behavior [37]. The impact of immature defense styles understood in this way is consistent with the aforementioned finding of Overcompensation mediatory effect [32]. Surprisingly, none of the studies included in the review proved the mediating role of parenting styles or parental attitudes in the intergenerational transmission of EMS [30,34]. Parenting styles reflect the way in which caregivers fulfill basic parenting responsibilities related to the child’s socialization, parental control, and discipline, and what emotional environment they create in the family [42]. This construct is strongly related to parental attitudes, which are defined as a set of knowledge, beliefs, values, attribution, expectations, and ideas about child-rearing [43]. Undoubtedly parenting styles and parental attitudes can express early maladaptive schemas of parent. This lack of confirmation of an indirect link between parenting attitudes/style and the transmission of EMSs may explain the fact that, unlike EMSs, parenting attitudes/styles may result from a more conscious decision to fulfill the parental role. On the other hand, an important factor that distorts the verification of these links may be the use of self-report and retrospective research methods, which may lead to wording or interpretation biases.

Interestingly, one study found that the emotional involvement of the father in the child’s caring process can mitigate the negative impact of maternal maladaptive schemas in the context of EMSs formation. The inverse relationship has not been studied. This highlights a significant gap in the research conducted so far related to ignoring the influence of fathers on the development of the maladaptive schemas of their children [36].

#### 3.5.2. The Impact of EMSs of Parent on Parenting

Onlytwo of the selected studies investigated the relationship between parents’ EMSs and their parental dispositions. The first study focused on the quality of maternal–fetal bonding and its association with mother’s EMSs domains, and thus far is the only one that addresses this co-morbidity. A study by Nordahl et al. confirmed the presence of significant correlations between all EMS domains and the quality of the prenatal bonding to the fetus. Based on regression analysis, the researchers indicated that a considerable part of the variance of bonding quality (32%) can be explained by the four EMS domains and seven potentially confounding factors (e.g., maternal age, education, parenting experience, and mental health history). EMS domains (except Excessive Responsibility and Standards) also showed a direct effect on bond quality when confounders and depressive symptoms were controlled. The most accurate predictor of the prenatal bonding quality was the Disconnection and Rejection domain. Moreover, the domain of Disconnection and Rejection plays a unique and significant role in estimating the intensity of fetal preoccupation, and depressive symptoms did not mediate this relationship. In line with previous research, the study also confirmed significant positive correlations between the intensity of EMSs and the severity of depressive symptoms [35]. The second of the selected studies verified the association between parent’s EMSs and their parental sense of competence. The results indicated that higher expression of EMSs was significantly and negatively correlated with parents’ perceptions of their own competence, regardless of the impact of the child’s behavioral problems, range of experiences in a care giving, or the level of education, which provides preliminary evidence supporting the hypothesis about the role of EMSs in sustaining negative parenting patterns [31].

The abbreviated form of the obtained results is presented in Table 3.

## 4. Strengths and Limitations

The results of this systematic review provide initial support for the conceptualization of multi-generational transmission of early maladaptive schemas. These preliminary findings are reinforced by a structured and methodical approach to data collection and extraction in line with PRISMA guidelines. The data collected from the studies selected in this review come from different populations of several continents, providing a general outline of the considered topic. Despite the contributions, some limitations should be mentioned. The first limitation to note is the small number of available studies that met the inclusion criteria, which illustrate a significant gap in the literature addressing this important topic. Comparative research, taking into account cultural and socio-demographic differences, is necessary to ensure a better understanding of the phenomenon under study. Second, reliance on cross-sectional research projects, as in selected studies, drastically limits the inference about cause-and-effect relationships. There is a great need for further longitudinal studies that will provide the possibility of establishing the causality and temporality of the mechanisms underlying the links between the EMSs of parents and their adult children. In addition, the vast majority of studies have been conducted on a non-clinical, well-functioning sample, in which the expression of EMS is relatively low. In future research, it may be helpful to extend the samples to clinical conditions. Third, all researchers used self-report questionnaires, which could lead to unreliable responses. Mixed assessment methods are recommended for further studies. Finally, in the studied populations, female gender was overwhelmingly predominant; some studies only took into account the mother–child relationship, while others used a mixed sample, still with a significant inequality between the sexes. It should be noted that there were differences in the character of EMSs transmission for parent–child dyads depending on their gender. It should be noted that in the parent–child dyads there were gender differences in the nature of EMSs transmission, therefore generalizing conclusions regarding the intergenerational transmission of schemas, without taking into account the gender of the sample, may lead to bias.

## 5. Conclusions: Future Research

The main objective of this systematic review was to evaluate evidence on the intergenerational transmission of EMSs, along with the underlying mechanisms. The obtained results indicate the emerging support for the relationship between the early maladaptive schemas of parents and the development of schemas in their children. However, due to the limited amount of data, it is not possible to make an unambiguous conclusion as to the nature of these associations. Further longitudinal studies with gender differentiated samples are highly recommended. To sum up, there are two main conclusions from these systematic reviews. First, the collected results consistently prove that the Disconnection/Rejection domain of parents has a negative effect on the parent–child bond, severely impairs parenthood, and thus facilitates schema transfer to the next generation (both direct and indirect). This finding is in line with previous research that has consistently identified this particular area as the most destructive, contributing to the development of a broad spectrum of mental health problems [44,45,46,47]. Secondly, recollections of adverse childhood experiences mediate the relationship between the EMSs of parent and EMSs of their children. We summarize these conclusions in the proposed model of the mediation of adverse childhood experiences on the relationship between parent’s disconnection and rejection schemas and the child’s disconnection and rejection schemas (Figure 2). Finally, the findings encourage approaches to minimize and prevent the negative effects of early maladaptive schemas along with their damaging effects on psychological well-being. 

## Figures and Tables

**Figure 1 jcm-12-01263-f001:**
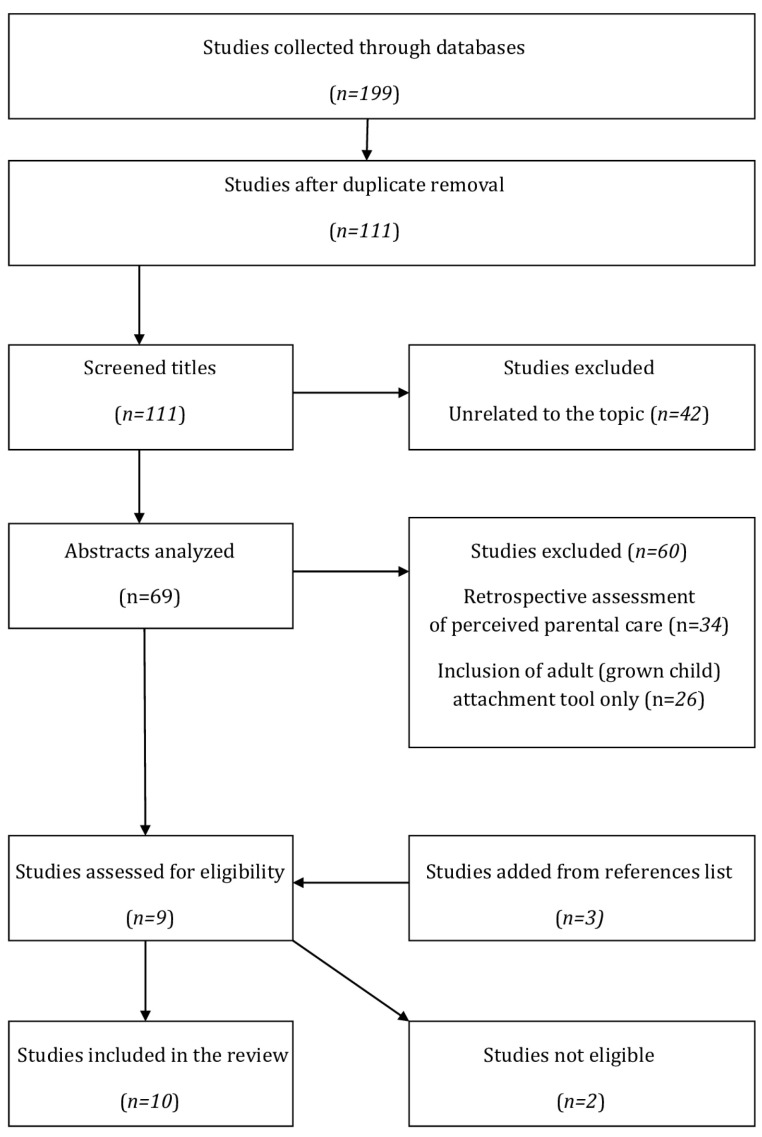
PRISMA (Preferred Reporting Items for Systematic Reviews and Meta-Analyses) flow chart of studies selection.

**Figure 2 jcm-12-01263-f002:**
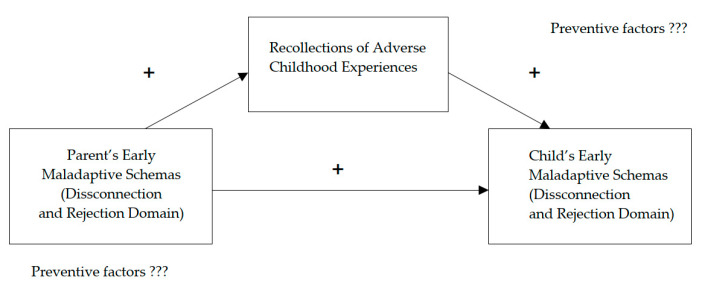
The proposed model of the mediation of recollections of adverse childhood experiences on the relationship between parent’s disconnection and rejection schemas and the child’s disconnection and rejection schemas.

**Table 1 jcm-12-01263-t001:** Short description of the early maladaptive schemas.

Disconnection and Rejection				
** *Emotional deprivation* **	** *Abandonment/Instability* **	** *Mistrust/Abuse* **	** *Social isolation/Alienation* **	** *Defectiveness/Shame* **
A conviction that basic emotional needs for nurturance, empathy, and protection cannot be adequately met by others	An expectation that relationships with others are unstable, insecure, fragile, and may end unexpectedly	Anticipating that others will intentionally harm, punish, humiliate, or take advantage	A feeling of a deep separation from society, not belonging to any community	A belief that one cannot be loved and accepted because of being flawed, inferior, bad, or imperfect
**Impaired autonomy and performance**				
** *Failure to achieve* **	** *Dependence/incompetence* **	** *Vulnerability to harm or illness* **		** *Enmeshment/undeveloped self* **
A belief that one does not have sufficient competences, talents, and intelligence to achieve results alike others in terms of career, education, and achievements.	A feeling of being completely helpless, powerless, unable to function independently	An expectation that the world is full of unpredictable catastrophes, threats, dangers, and a person has no resources to deal with it		Excessive, emotional involvement in the life of a loved one (s), associated with a sense of identity fusion or blurring, and the fear that one person cannot survive without the constant devotion to the other
**Impaired limits**				
** *Entitlement/grandiosity* **		** *Insufficient self-control/self-discipline* **		
A belief in one’s own superiority over others, having special privileges or being above the applicable laws and rules		Recurring difficulties with self-control, emotional management, frustration tolerance, deferring gratification		
**Other directedness**				
** *Subjugation* **		** *Self-sacrifice* **		** *Approval-seeking/Recognition-seeking* **
The belief that one must submit to the will of others in order to avoid negative consequences (e.g., punishment, conflict, rejection)		The conviction that satisfying the needs of others should be inviolably placed above one’s own		Excessive concentration on attention, acceptance and appreciation of the social environment, on which a person depends for self-esteem
**Over-vigilance and inhibition**				
** *Emotional inhibition* **	** *Unrelenting standards/hypercriticalness* **	** *Negativity/pessimism* **		** *Punitiveness* **
An absolute imperative to keep control over one’s feelings, emotional reactions, and impulses, in order to avoid feelings of shame, rejection, disapproval	A belief that, regardless of the efforts put in, one will never be good enough and never live up to expectations, which results in rigid behavior, perfectionism, and denying oneself the pleasure	A perception of life through the prism of negative aspects, deficiencies, flaws, and minimizing its positive aspects, often combined with a tendency to worry		A belief that people should be severely punished for their mistakes combined with an attitude of intolerance, inexcusability

**Table 2 jcm-12-01263-t002:** Characteristics of the Studies.

Year/ Location	Author	Sample Size	Characteristic of Participants	Objective	Assessment Measures *
2012, USA	Shorey, Anderson, Stuart [29]	Total: 105	The total number of participants was 105. Substance abuse treatment-seeking adults: Males = 32, Females = 15 Age: M = 29.63, SD = 9.57 Parents: Mothers = 13, Fathers = 45 Age: M = 58.13, SD = 8.66	To consider similarities and differences in EMSs among a sample of substance abuse treatment-seeking adults and at least one parent.	▪ Young Schema Questionnaire—Long Form, Third Edition (YSQ-L3) (GCh/P) ▪ The Alcohol Use Disorders Identification Test (AUDIT) (GCh/P) ▪ The Drug Use Disorders Identification Test (DUDIT) (GCh/P)
2016, Poland	Mącik, Chodkiewicz, Bielicka [30]	Total: 80	The total number of participants was 20 full families with grown children: a daughter and a son. Grown children: Age: M = 27.83, SD = 3.26 Parents: Age: M = 53.83, SD = 4.28	To explore the relations between dysfunctional parents’ EMSs and their parental attitudes and their children’ EMSs.	▪ Retrospective Assessment of Parents’ Attitudes (KPR-Roc) (GCh) ▪ The Young Schema Questionnaire Short Form-3 (YSQ-SF3) (GCh/P)
2017, Hungary	Miklósi, Szabó, Simon [31]	Total: 145	The total number of participants was 145 caregivers, 1 excluded due to incomplete data. There were 122 mothers, and 19 fathers, and 3 other caregivers. Children: Males = 48, Females = 96 Age: M = 10.58, SD = 5.50 Parents: Age: M = 40.36, SD = 6.65	To test the associations between parents’ perceptions of their own aversive childhood experiences with their caregivers, the extent of their EMSs, and their current level of perceived parenting competence. Secondly, to explore whether parents’ level of mindfulness is moderating or mediating this relationship.	▪ Strengths and Difficulties Questionnaire (SDQ) (P) ▪ Young Parenting Inventory (YPI)(P) ▪ The Young Schema Questionnaire Short Form-3 (YSQ-SF3) (P) ▪ Mindful Attention Awareness Scale (MAAS)(P) ▪ The Parental Sense of Competence Scale (PSOC)(P)
2018, Germany	Sundag, et al. [32]	Total 120	The total number of participants was 60 parent–adult child dyads. Grown children: Daughter =38, Son = 22 Age: M = 28.4, SD = 9.4 Parents: Mothers = 49, Fathers = 11 Age: M = 57.8, SD = 8.7	To investigate whether the extent of EMSs in parents is associated with the extent of EMSs in their offspring.	▪ Young Schema Questionnaire, Short Form 3 Revised (YSQ-S3R) (GCh/P) ▪ Young Parenting Inventory (YPI) (GCh) ▪ Young Compensation Inventory(YCI) (P) ▪ Young–Rygh Avoidance Inventory (YRAI-1)(P)
2018, Netherlands	Zonnevijlle, Hildebrand [33]	Total: 40	The total number of participants was 20 parent–adolescent dyads. Adolescents: Males = 13, Females = 7 Age: M = 16.2 years, SD = 1.6 Parents: Mothers = 19, Fathers = 1 Age: M = 45.6 years, SD = 8.1	To examine the interrelationships of and differences between EMSs among maltreated children and their parents.	▪ Young Schema Questionnaire-Short Form-3 (YSQ-SF3) (A/P) ▪ EgnaMinnenBeträffandeUppfostran for Children (EMBU-C) (A) ▪ Experiences in Close Relationships Scale–Revised (ECR-RC) (A)
2019, Australia	Gibson, Francis [34]	Total: 100	The total number of participants was 100. There were 43 mothers and 57 adult daughters; 41 were matched in dyads and 39 included in the analysis. Adult daughters: Age: M = 26.28, SD = 9.33 Mothers: Age: M = 55.74, SD = 8.75	To verify the potential mediating role of parenting styles in relationships between mothers’ and daughters’ EMSs.	▪ The Young Schema Questionnaire Short Form-3 (YSQ-SF3) (M/AD) ▪ Depression Anxiety and Stress Scale Short Form (DASS-21) (M/AD) ▪ The Parental Authority Questionnaire (PAQ) (AD) ▪ Parental Bonding Instrument (PBI) (AD)
2019, Norway	Nordahl, et al. [35]	Total: 165	The total number of participants was 165 pregnant women. Age: M = 30.8, SD = 4.1	To examine the relationship between mothers’ EMSs and two aspects of maternal–fetal bonding: the intensity of preoccupation with the fetus and the quality of the affective bond.	▪ Young Schema Questionnaire Short Form 3 (YSQ-S3) (PW) ▪ The Maternal Antenatal Attachment Scale (MAAS) (PW) ▪ The Edinburgh Postnatal Depression Scale (EPDS) (PW)
2020, Turkey	Zeynel, Uzer [36]	Total: 358	The total number of participants was 179 mother–late adolescent dyads. Late adolescent: Males = 83, Females = 95, Unwilling to report = 1 Age: M = 20.52, SD = 1.16 Mothers = 179 Age: M = 47.64, SD = 5.31	To test the mechanisms underlying the relationship between the parent’s disconnection and rejection schemas and the child’s disconnection and rejection schemas.	▪ Young Schema Questionnaire-Short Form-3 (YSQ-SF3) (LA/M) ▪ Childhood Trauma Questionnaire (CTQ)(LA) ▪ Father Involvement Scale (FIS)(LA) ▪ Resilience Scale (RS) (A)
2021, Turkey	Karaarslan, Eldogan, Yigit [37]	Total: 626	The total number of participants was 215 families (i.e., mother, father, and their adult children) Adolescents: Males = 66, Females = 149 Age: M = 20.82, SD = 2.71 Mothers = 201 Age: M = 48.38, SD = 5.15 Fathers = 210 M = 52.08, SD = 5.55	To evaluate the mediating role of defense styles in the associations between two EMS domains (Disconnection/Rejection and Impaired Autonomy) of parents and their adult children.	▪ Young Schema Questionnaire-Short Form (YSQ-SF) (A/P) ▪ Defence Style Questionnaire (DSQ) (A/P)
2022, Turkey	Alaftar, Uzer [38]	Total: 240	The total number of participants was 120 mother–late adolescent dyads. Adolescents Age: M = 21.78, SD = 1.50 Mothers Age: M: 49.93, SD = 4.56	To examine whether overgeneral autobiographical memory facilitates the transmission of early maladaptive schemas (EMSs) by strengthening maladaptive thinking patterns after traumatic experiences.	▪ Autobiographical Memory Recall Task (A) ▪ The Young Schema Questionnaire Short Form-3 (YSQ-SF3) (A/M) ▪ Beck Depression Inventory (BDI) (A/M) ▪ Childhood Trauma Questionnaire (CTQ) (A)

* Assessment measures—to whom the questionnaire was addressed—is given in brackets, (A)—adolescents, (M)—mothers, (AD)—adult daughters, (P)—parents, (PW)—pregnant women, (LA)—late adolescents.

**Table 3 jcm-12-01263-t003:** Main Results of the Studies included.

Author, Year	Main Results and Conclusions
2012 Shorey, Anderson, Stuart [29]	Lack of support for the hypothesis of the intergenerational transmission of early maladaptive schemas until parents showed high scores for most EMSs. Significantly higher scores in 17 out of 18 EMSs (early maladaptive schemas) of substance abusers seeking treatment than their parents.
2016 Mącik, Chodkiewicz, Bielicka [30]	Support for the hypothesis that early maladaptive schemas may be transmitted intergenerationally, however not in a straight way. Children’s EMSs become the answer to parents’ EMSs, in the case of daughters, more complementary, in the case of sons, the reverse.
2017 Miklósi, Szabó, Simon [31]	Self-reports of childhood neglect are significantly related to higher EMSs scores and correlate with the caregivers’ current sense of competence in their own parenting roles. Higher intensity of EMSs is related to lower levels of mindfulness, and consequently with lower levels of parental competence.
2018 Sundag, et al. [32]	The extent of parents’ EMSs is a significant predictor of the intensity of a child’s EMSs. The parental schema coping style of Overcompensation and the adverse parenting that the child remembered are assumed to underlie the intergenerational transmission of EMSs.
2018 Zonnevijlle, Hildebrand [33]	Unrelenting standards schema demonstrate significant positive association between parents and youth scores. There are substantial correlations between parents’ schemas of the Impaired limits and Disconnection/rejection domains and children’s schemas of the Disconnection/rejection and Impaired autonomy and performance domains.
2019 Gibson, Francis [34]	Daughters’ schemas of Subjugation and Approval seeking are most strongly associated with overall mothers’ schemas. Mothers’ schemas in the Disconnection/Rejection domain are significantly related to daughters’ overall schema scores. An abandonment and Mistrust/Abuse schema of mothers and daughters are directly related.
2019 Nordahl, et al. [35]	Significant, negative correlation between all domains of EMSs and the quality of the maternal–fetal bonding. The Disconnection and Rejection domain as a significant, independent predictor of the quality of maternal–fetal bonding. Depressive symptoms mediate the effect between pregnant women’s EMSs domains (Disconnection and Rejection, Impaired Autonomy and Performance, and Impaired Limits) and the quality of the maternal–fetal bond.
2020 Zeynel, Uzer [36]	Mediating role of adverse childhood experiences in the relationship between mother’s Disconnection and Rejection domain schemas and her child’s Disconnection and Rejection domain schemas. Protective role of fathers involvement in childcare against intergenerational transmission of EMSs.
2021 Karaarslan, Eldogan, Yigit [37]	Significant correlations between parents’ and their adult children’s Disconnection and Rejection and Impaired Autonomy EMSs domains were found. Mediating influence of immature defense styles of parents and their adult children.
2022 Alaftar, Uzer [38]	Support for hypothesis that adverse childhood experiences significantly mediated the relationship between mothers’ and children’s disconnection and rejection schemas. Overgeneral autobiographical memory can intensify the association between adverse childhood experiences and children’s Disconnection and Rejection schemas.

## Data Availability

Data sharing is not applicable to this article as no new data were created in this study.

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
