# Peer review of "Early Maladaptive Schemas and Their Impact on Parenting: Do Dysfunctional Schemas Pass Generationally?—A Systematic Review"

_jcm, 2023, doi:10.3390/jcm12041263_

Round 1

Author Response

Dear Reviewer,

We sincerely thank you for your in-depth review of our manuscript and for the excellent suggestions we have received. We have made a coordinated effort to adequately respond to every suggestion we receive. We strongly believe that your comments and suggestions have greatly improved this manuscript.

Below we provide the point-by-point responses. All modifications in the manuscript have been highlighted in red.

Point 1: The title is a tad too long.

Response: Early maladaptive schemas and their impact on parenting. Do dysfunctional schemas pass generationally? - A Systematic Review

Point 2: The first part of the abstract does not describe EMS well: “Adverse childhood have a profound impact on lifespan health and well-being. It has been repeatedly noticed that children of parents, who were exposed to abusive behaviors from their own parents, are significantly more likely to experience domestic violence, which is called “perpetuation across generations”. Maternal mental health problems have been proven to be a factor that has strengthened the link between a mother's history of negative childhood experiences and subsequent negative parenting.” We cannot assume that adversities equal to negative EMS Please revise the abstract, which needs to be linked with the introduction part as well.

Response: We revised the abstract as follows:

Thus, the parental care that a child experiences has a substantial impact on the potential development of early maladaptive schemas. Negative parenting can range from unconscious neglect to overt abuse. Previous research supports the theoretical concept that there is a clear and close relationship between adverse childhood experiences and the development of early maladaptive schemas. (Pg1, Ln 28-32).

Point 3: First paragraph line 35 – The term malparenting deserves a bit of introduction here.Does it cover negative or punitive parenting? Is there any specific parenting style associated with it?

Response: We clarify the concept:

malparenting (ranging from unconscious neglect to overt abuse) (Pg1, Ln 52).

Point 4: Line 42-45: In-text citation from Young et al should be provided with the exact pages (pp.) the word-to-word citation is taken from.

Response: Revised accordingly (Pg16, Ln 28)

Point 5: Line 64-79: The authors have earlier justified how some psychopathology/personality disorders development is mediated and sustained by EMSs. However, the relationship between EMSs and child abuse is not clearly described here. While it is acknowledged that parental adverse childhood might affect their child through parenting, whether this occurs through EMSs or not, is not clear.

Response: We pointed to the lack of knowledge about the mechanisms underlying the transmission of adverse childhood experiences. 

Although the relationship between adverse childhood experiences and early maladaptive schemas is well established in research, little is still known about the role of EMS in perpetuating negative parenting across generations. (Pg 3, Ln 50-52). 

Point 6: I would also appreciate if Table 3 could be made simpler.

Response: Revised accordingly (Pg13, Ln 31).

Point 7: Table 1 and Figure 2 were not mentioned in the text.

Response: Revised accordingly, Tab 1  (Pg 2, Ln 8),  Fig 2 (Pg 15, Ln 30)

Point 8: This is slightly long. Please state the conclusive statements only. 

Response: We shortened the section by removing suggestions for therapeutic solutions. (Pg 15, Ln 13-32)

We enclose the revised manuscript in the attachment below.

Best regards,

Klaudia Sójta

Reviewer 2 Report

Overall, this is a well-written, original and important review on the transmission of early maladaptive scheme's of parents towards their children in a mainly non-clinical population.

I hav some minor points for improvement:

Line 116-117: What do the authors mean in the manuscript (MS) by the inclusion criterium: “IV) adaptation of a case–control, longitudinal, cross-sectional or 116 retrospective study design?” Figure 1 states that “retrospective assessment of parenting" has been used as an exclusion criterium, excluding 34 studies, and leaving only 9. This might be a sound decision, but is not elaborated. What exactly is the difference between these 34 and 9 paper? If these exclusions are meant to be, why? Please explain in the methods.

Line 165: “Simultaneously, we created a textual description of each study selected, subjecting it to a critical evaluation”. What do the authors mean by a critical evaluation, what process or criteria do they use? Please clarify in the MS.

Line 166-167: “Then, we explored the relationships within and between the selected studies, using a grouping technique”. What do the authors mean by ‘grouping technique, please explain in the MS.

Line 184: The remark: “two were excluded due to failure to meet the eligibility criteria”, is not added to figure 1, please add. 

Author Response

Dear Reviewer,

We sincerely thank you for your in-depth review of our manuscript and for the excellent suggestions we have received. We have made a coordinated effort to adequately respond to every suggestion we receive. We strongly believe that your comments and suggestions have greatly improved this manuscript.

Below we provide the point-by-point responses. All modifications in the manuscript have been highlighted in red.

Point 1: Line 116-117: What do the authors mean in the manuscript (MS) by the inclusion criterium: “IV) adaptation of a case–control, longitudinal, cross-sectional or 116 retrospective study design?” Figure 1 states that “retrospective assessment of parenting" has been used as an exclusion criterium, excluding 34 studies, and leaving only 9. This might be a sound decision, but is not elaborated. What exactly is the difference between these 34 and 9 paper? If these exclusions are meant to be, why? Please explain in the methods.

(III) focusing on associations between early maladaptive schema of parent and various aspects of parenting (e.g. parental attitudes, sense of parenting competences, parenting styles, etc.) (Pg 5, Ln 4-6).

Considering only retrospective assessment of perceived parental care  (relate to the child's perception of parents )(n=34) (Pg 6, Ln 21-22)

Point 2: Line 165: “Simultaneously, we created a textual description of each study selected, subjecting it to a critical evaluation”. What do the authors mean by a critical evaluation, what process or criteria do they use? Please clarify in the MS.

Response: We defined the evaluation criteria

Simultaneously, we separately created a textual description of each selected study, assigning importance ranks based on methodological quality criteria (Pg 6, Ln 2-4).

Point 3: Line 166-167: “Then, we explored the relationships within and between the selected studies, using a grouping technique”. What do the authors mean by ‘grouping technique, please explain in the MS.

Response: We described the use of the grouping technique in detail.

We then divided the included studies into groups to explore relationships within and between the selected studies. The studies were grouped according to the following criteria: target group (e.g. parent, parent-child dyad); outcome measures, various aspects of EMS impact (e.g. bonding, parental sense of competence). (Pg 6, Ln 4-8)

Point 4: Line 184: The remark: “two were excluded due to failure to meet the eligibility criteria”, is not added to figure 1, please add.

Response: We added an item with excluded studies. (Figure 1, Pg 4, Ln 22)

We enclose the revised manuscript in the attachment below.

Best regards,

Klaudia Sójta
